# Multilingual *k*-Nearest-Neighbor Machine Translation

**David Stap**    **Christof Monz**
Language Technology Lab
University of Amsterdam
{d.stap, c.monz}@uva.nl

## Abstract

*k*-nearest-neighbor machine translation has demonstrated remarkable improvements in machine translation quality by creating a datastore of cached examples. However, these improvements have been limited to high-resource language pairs, with large datastores, and remain a challenge for low-resource languages. In this paper, we address this issue by combining representations from multiple languages into a single datastore. Our results consistently demonstrate substantial improvements not only in low-resource translation quality (up to $+3.6$ BLEU), but also for high-resource translation quality (up to $+0.5$ BLEU). Our experiments show that it is possible to create multilingual datastores that are a quarter of the size, achieving a 5.3x speed improvement, by using linguistic similarities for datastore creation.[1]

## 1 Introduction

Recently, semi-parametric approaches such as *k*-nearest-neighbor machine translation (*k*NN-MT) (Khandelwal et al., 2021) have attracted interest due to a series of impressive results in language modeling and machine translation (Guu et al., 2018; Bapna and Firat, 2019; Khandelwal et al., 2020). These techniques capitalize on information retrieved from an extensive repository of translation examples cached in a datastore. One of the most important limitations of *k*NN-MT is that the extent of quality improvements strongly depends on the size of the datastore (Khandelwal et al., 2020, 2021; Zhu et al., 2023). This dependence on datastore size is problematic for low-resource languages and the improvements that *k*NN-MT can offer in low-resource settings are modest at best (Vardhan et al., 2022). On the other hand, there are general methods that can improve low-resource performance, such as transfer learning (Zoph et al., 2016; Kocmi

---

and Bojar, 2018) and multilingual NMT (mNMT) (Johnson et al., 2017; Arivazhagan et al., 2019; Stap et al., 2023).

Preliminary findings on combining *k*NN-MT with mNMT suggest that mNMT representations generalize sufficiently well across languages to make cross-lingual retrieval effective (Khandelwal et al., 2021). However, its effectiveness for low-resource languages remains an open question.

In this paper, we investigate to what extent mNMT can be useful for improving low-resource *k*NN-MT translation quality. First, we experiment with cross-lingual datastores from related languages and find that low-resource languages generally benefit from larger cross-lingual datastores. We then propose a simple yet effective approach, *multilingual* *k*NN-MT, which uses multilingual datastores that are constructed by merging bilingual datastores. Our results show substantial improvements for low-resource languages, and also noticeable improvements for high-resource languages. Finally, we show that it is possible to create multilingual datastores that are significantly smaller—thereby resulting in substantially faster decoding times—by relying on linguistic similarities when creating multilingual datastores.

## 2 *k*-nearest neighbor machine translation

*k*NN-MT combines a parametric component with a nearest neighbor retrieval mechanism that allows direct access to a datastore of cached examples (Khandelwal et al., 2021). The datastore $\mathcal{D}$ consists of key-value pairs, where each key is a *translation context*, i.e., decoder output representation, $f(\mathbf{x}, \mathbf{y}_{<t})$, and the value is the corresponding target token $y_t$. At inference time, the model searches the datastore to retrieve the set of $k$ nearest neighbors $\mathcal{N}$. Using their distances $d(\cdot)$ to the current translation context, a retrieval distribution $p_{k\text{NN}}(y_t|\mathbf{y}_{<t}, \mathbf{x})$ is computed. The final probability distribution is obtained by combining

---

[1]We release our code at https://github.com/davidstap/multilingual-kNN-mt.

$p_{\text{NMT}}(y_t|\mathbf{y}_{<t}, \mathbf{x})$ and $p_{k\text{NN}}(y_t|\mathbf{y}_{<t}, \mathbf{x})$.

## 3 Multilingual *k*-nearest-neighbor MT

Despite the potential benefits, the integration of mNMT with *k*NN-MT has only been rarely explored. Using a datastore with English on the source side improves performance for other source languages (Khandelwal et al., 2021), but it is not known to what extent this holds for low-resource languages. Pre-trained multilingual language models can be used to build monolingual datastores of a target language (Li et al., 2022), but the required alignment training does not work for low-resource languages due to data scarcity. Our goal is to improve performance for low-resource languages by constructing cross-lingual and multilingual datastores. These datastores consist of keys generated from mNMT representations, allowing semantically related sentences from different languages to cluster together (Johnson et al., 2017; Escolano et al., 2019).

### 3.1 Bilingual and cross-lingual datastores

A *bilingual datastore* is defined as follows:

$$\mathcal{D}_{(\ell,\ell')} = \{(f(\mathbf{x}, \mathbf{y}_{<t}), y_t), \forall y_t \in \mathbf{y} \mid (\mathbf{x}, \mathbf{y} \in \mathcal{B}_{(\ell,\ell')})\}, \quad (1)$$

where bi-text data $\mathcal{B}_{(\ell,\ell')}$ originates from a single source language $\ell$ into target language $\ell'$. When we use a bilingual datastore $\mathcal{D}_{(\ell,\ell')}$ to augment the translation direction of another source language $\ell^* \neq \ell$ into target language $\ell'$, we call the datastore *cross-lingual*. For instance, a Russian-English datastore $\mathcal{D}_{(\text{ru,en})}$ may be used to enhance Belarusian-English translation. An important advantage of cross-lingual datastores is that they can be significantly larger than their bilingual counterparts, and therefore may result in better translation quality.

### 3.2 Multilingual datastores

Earlier work is limited to monolingual or bilingual datastores (Khandelwal et al., 2021; Cai et al., 2021; Li et al., 2022). In contrast, we create multilingual datastores consisting of multiple source languages, resulting in larger datastores.

We construct a *multilingual datastore* $\mathcal{D}_{(\text{L}_{\text{ML}},\ell')}$ by considering a set of source languages $\text{L}_{\text{ML}}$ that map to a target language $\ell'$:

$$\mathcal{D}_{(\text{L}_{\text{ML}},\ell')} = \{(f(\mathbf{x}, \mathbf{y}_{<t}), y_t), \forall y_t \in \mathbf{y} \mid (\mathbf{x}, \mathbf{y}) \in \mathcal{B}_{(\text{L}_{\text{ML}},\ell')}\}, \quad (2)$$

where $\mathcal{B}_{(\text{L}_{\text{ML}},\ell')} = \bigcup_{\ell \in \text{L}_{\text{ML}}} \mathcal{B}_{(\ell,\ell')}$ is the combined data from all $\text{L}_{\text{ML}}$ source languages into target $\ell'$.

To further align multilingual representations, we learn a linear mapping between two languages. Our goal is to let language $\ell^1$ more effectively query from a $\ell^2$ datastore. For our training data $\mathbb{T}$, we include translation contexts from the $\ell^1$ datastore $\mathcal{D}_{(\ell^1,\ell')}$ and the $\ell^2$ datastore $\mathcal{D}_{(\ell^2,\ell')}$ that correspond to the *same* target sentence $\mathbf{y}$ and target token $y_t$:

$$\mathbb{T} = \{(\mathcal{D}^i_{(\ell^1,\ell')}, \mathcal{D}^j_{(\ell^2,\ell')}) \mid i, j \text{ map to } y_t \in (\mathbf{y} \in \mathcal{B}_{(L_{12},\ell')})\}, \quad (3)$$

where $\mathcal{B}_{(L_{12},\ell')} = \{\mathcal{B}_{(\ell^1,\ell')}, \mathcal{B}_{(\ell^2,\ell')}\}$ is the combined data from $\ell^1$ and $\ell^2$ into $\ell'$. Subsequently, we minimize $\min_A \sum_{i=1}^n = ||\mathbb{T}^i_{\ell^2} - A\mathbb{T}^i_{\ell^1}||$ using the normal equation approach, where $\mathbb{T}^i_{\ell^1}$ and $\mathbb{T}^i_{\ell^2}$ originate from tuples of $\mathbb{T}$. We then use $A$ to map a translation context from $\ell^1$ to $\ell^2$. We also learn the inverse relation, i.e., $\ell^2$ to $\ell^1$, and create an optimized multilingual datastore for $\ell^1$ by applying the $\ell^2$ to $\ell^1$ mapping prior to storing the datastore.

## 4 Experiments

### 4.1 Setup

The 418M parameter version of the M2M100 multilingual translation model (Fan et al., 2021) is used for all experiments. It is a Transformer (Vaswani et al., 2017) with 24 layers, 16 heads and hidden dimensionality of 1024 supporting 100 languages.

We conduct our experiments on the widely used TED Talks corpus (Qi et al., 2018). We use the train set to create datastores, the development set for tuning *k*NN-MT hyperparameters, and the test set to report results. We use 51 languages into English, from 23 different language families, that are supported by both TED and M2M100.

We use *k*NN-BOX (Zhu et al., 2023) for our experiments. Following Khandelwal et al. (2021), we tune the number of neighbors $k \in \{16, 32, 64\}$, interpolation $\lambda \in \{0.2, 0.3, ..., 0.7\}$ and softmax temperature $T \in \{10, 100\}$ hyperparameters on the development set. We use a beam size of 5. We evaluate our models using sacreBLEU (Post, 2018; Papineni et al., 2002).[2]

### 4.2 Cross-lingual and multilingual datastores

We construct datastores for languages from three M2M100 language groupings into English: Slavic (12 languages), Germanic (5 languages), and Greek (4 languages). We then generate translations for all possible combinations of source language and datastore, with the goal of investigating the potential of

---

[2] nrefs:1|case:mixed|eff:no|tok:13a|smooth:exp|version:2.3.1

Table 1 (Slavic, top):

| $\mathcal{D}$ | base | $\mathcal{D}_{be}$ | $\mathcal{D}_{bs}$ | $\mathcal{D}_{sl}$ | $\mathcal{D}_{mk}$ | $\mathcal{D}_{sk}$ | $\mathcal{D}_{cs}$ | $\mathcal{D}_{uk}$ | $\mathcal{D}_{hr}$ | $\mathcal{D}_{sr}$ | $\mathcal{D}_{bg}$ | $\mathcal{D}_{pl}$ | $\mathcal{D}_{ru}$ | $\mathcal{D}_{LG}$ | $\mathcal{D}_{BR}$ | $\mathcal{D}_{ALL}$ |
|---|---|---|---|---|---|---|---|---|---|---|---|---|---|---|---|---|
| $|\mathcal{D}|$ | 0 | 116K | 146K | 520K | 683K | 1.6M | 2.7M | 2.9M | 3.3M | 3.6M | 4.7M | 4.7M | 5.6M | 30.6M | 86.4M | 125M |
| be | 19.2 | 20.9 | 20.5 | 20.9 | 20.8 | 21.0 | 21.5 | 22.4 | 21.4 | 21.3 | 21.3 | 21.3 | 21.7 | 23.1 | 22.2 | 22.5 |
| bs | 31.5 | 33.2 | 33.1 | 34.0 | 34.0 | 34.4 | 34.6 | 34.7 | 36.0 | 35.8 | 35.2 | 34.6 | 35.0 | 36.7 | 36.0 | 36.2 |
| sl | 24.9 | 26.4 | 26.4 | 27.3 | 27.0 | 27.6 | 27.9 | 27.9 | 28.3 | 28.4 | 27.9 | 28.1 | 28.2 | 29.2 | 29.2 | 29.5 |
| mk | 29.3 | 32.0 | 32.1 | 32.5 | 32.8 | 33.2 | 33.8 | 33.3 | 34.8 | 33.9 | 34.1 | 33.4 | 33.4 | 35.5 | 35.5 | 35.6 |
| sk | 28.4 | 30.3 | 30.5 | 31.5 | 31.4 | 32.6 | 32.1 | 32.2 | 32.4 | 32.8 | 32.4 | 32.5 | 34.1 | 34.1 | | |
| cs | 27.5 | 29.3 | 29.4 | 30.0 | 30.0 | 30.9 | 31.4 | 30.7 | 30.8 | 30.9 | 31.2 | 30.8 | 31.0 | 32.0 | 31.9 | 32.1 |
| uk | 24.7 | 26.6 | 27.0 | 27.4 | 27.6 | 27.9 | 28.3 | 29.1 | 28.5 | 28.3 | 28.8 | 28.3 | 28.9 | 29.9 | 29.6 | 29.7 |
| hr | 32.2 | 33.8 | 34.4 | 34.8 | 34.9 | 35.3 | 35.6 | 35.5 | 37.0 | 36.6 | 36.0 | 35.5 | 35.7 | 37.5 | 37.1 | 37.8 |
| sr | 30.7 | 32.2 | 32.7 | 33.3 | 33.6 | 33.8 | 34.2 | 34.0 | 35.2 | 35.9 | 34.8 | 34.3 | 34.6 | 36.3 | 35.7 | 36.5 |
| bg | 34.4 | 36.1 | 36.2 | 37.1 | 37.2 | 37.4 | 37.9 | 37.6 | 38.1 | 38.2 | 39.5 | 38.0 | 38.3 | 39.7 | 39.5 | 39.9 |
| pl | 21.1 | 22.6 | 22.8 | 23.4 | 23.5 | 23.8 | 24.1 | 23.9 | 24.0 | 24.1 | 24.5 | 25.0 | 24.3 | 25.4 | 25.4 | 25.4 |
| ru | 21.6 | 23.3 | 23.3 | 23.8 | 24.1 | 24.3 | 24.7 | 24.9 | 24.7 | 24.8 | 25.0 | 24.9 | 25.8 | 26.0 | 26.0 | 25.4 |
| avg | 27.1 | 28.9 | 29.0 | 29.7 | 29.7 | 30.2 | 30.6 | 30.5 | 30.9 | 30.9 | 30.9 | 30.6 | 30.8 | 32.1 | 31.8 | 32.1 |

Table 1 (Germanic, bottom left):

| $\mathcal{D}$ | base | $\mathcal{D}_{no}$ | $\mathcal{D}_{da}$ | $\mathcal{D}_{sv}$ | $\mathcal{D}_{de}$ | $\mathcal{D}_{nl}$ | $\mathcal{D}_{LG}$ | $\mathcal{D}_{BR}$ | $\mathcal{D}_{ALL}$ |
|---|---|---|---|---|---|---|---|---|---|
| $|\mathcal{D}|$ | 0 | 411K | 1.2M | 1.4M | 4.5M | 4.9M | 12M | 86.4M | 125M |
| no | 42.8 | 45.6 | 46.7 | 45.9 | 46.4 | 46.3 | 47.7 | 47.4 | 47.8 |
| da | 40.0 | 43.1 | 44.5 | 43.3 | 44.0 | 44.0 | 45.5 | 45.0 | 45.7 |
| sv | 37.3 | 39.6 | 40.4 | 41.0 | 40.8 | 40.8 | 41.8 | 42.1 | 42.0 |
| de | 31.7 | 34.3 | 34.9 | 35.0 | 36.9 | 36.0 | 37.1 | 37.2 | 37.3 |
| nl | 31.9 | 33.9 | 34.4 | 34.6 | 35.2 | 36.2 | 36.1 | 36.0 | 36.3 |
| avg | 36.7 | 39.3 | 40.2 | 40.0 | 40.7 | 40.7 | 41.7 | 41.5 | 41.8 |

Table 1 (Greek, bottom right):

| $\mathcal{D}$ size | base | $\mathcal{D}_{ka}$ | $\mathcal{D}_{hy}$ | $\mathcal{D}_{sq}$ | $\mathcal{D}_{el}$ | $\mathcal{D}_{LG}$ | $\mathcal{D}_{BR}$ | $\mathcal{D}_{ALL}$ |
|---|---|---|---|---|---|---|---|---|
| | 0 | 332K | 544K | 1.2M | 3.5M | 5.6M | 86.4M | 125M |
| ka | 10.8 | 14.7 | 12.7 | 12.8 | 12.9 | 15.2 | 14.4 | 15.2 |
| hy | 16.8 | 18.4 | 20.1 | 19.0 | 19.4 | 20.8 | 20.3 | 20.7 |
| sq | 31.9 | 33.2 | 33.4 | 35.8 | 34.6 | 36.0 | 35.5 | 36.2 |
| el | 32.6 | 34.8 | 35.3 | 35.8 | 38.3 | 38.3 | 38.7 | 38.8 |
| avg | 23.0 | 25.3 | 25.4 | 25.9 | 26.3 | 27.6 | 27.2 | 27.7 |

Table 1: X→en BLEU scores for three language groupings: Slavic (top), Germanic (bottom left) and Greek (bottom right). We display results for all combinations of translation directions and datastores $\mathcal{D}$ within each language grouping. (For brevity, we write e.g. the Belarusian-English datastores as $\mathcal{D}_{be}$ instead of $\mathcal{D}_{(be,en)}$.) Datastore size is depicted as $|\mathcal{D}|$. We refer to languages for which $|\mathcal{D}| < 1M$ as low-resource languages. These languages are separated from high-resource languages by a dashed line. We color the bilingual datastores on the diagonal grey. The three rightmost columns are multilingual datastores, built from language grouping languages ($\mathcal{D}_{LG}$), bridge languages ($\mathcal{D}_{BR}$), or all languages ($\mathcal{D}_{ALL}$). BLEU scores obtained without kNN-MT are listed in the column labeled base. We underline the best cross-lingual or bilingual results. Overal best scores are depicted in **bold**.

cross-lingual datastores to improve low-resource performance.

Additionally, we construct several multilingual datastores:

$\mathcal{D}_{(ALL,en)}$: We created a comprehensive datastore, integrating 51 languages that occur in both TED and M2M100 into English, resulting in 125M entries.

$\mathcal{D}_{(BR,en)}$: mNMT is typically English-centric, i.e., English occurs on the source or target side in the training data. M2M100 instead uses a set of *bridge languages*, which leads to a greater coverage of direct translation directions. To align with these languages, we additionally create a smaller datastore of size 86.4M, consisting of 24 bridge languages.

$\mathcal{D}_{(LG,en)}$: We investigate to what extent datastore size can be further decreased. We hypothesize that more similar multilingual representations result in better cross-lingual retrieval, and that representations within the same language grouping are more similar. In line with this, we create three multilingual datastores consisting of all languages within a language grouping: Slavic (datastore size 30.6M), Germanic (datastore size 20M), and Greek (datastore size 5.6M).

See Appendix A for more details on the datastores.

## 4.3 Results

Translation results for bilingual, cross-lingual, and multilingual datastores are shown in Table 1.

**Bilingual datastores work to a limited extent** We observe that bilingual datastores can bring limited improvements in performance for low-resource languages, even though their datastores are small. For instance, low-resource languages from the Slavic language grouping improve with +2.3 BLEU on average. High-resource languages have more improvements, e.g., the Slavic grouping gains +4.5 BLEU on average.

**Cross-lingual is better than bilingual** In general, low-resource languages benefit from cross-lingual datastores. For instance, Belarusian-English (be-en) with bilingual datastore (116K instances) results in +1.7 BLEU, whereas Belarusian-English with a substantially larger Ukrainian-English (uk-en) datastore (2.9M instances) leads to a further improvement of +1.5 BLEU. However, while datastore size and performance do correlate[3], the additional quality improvements can *not* be fully explained by the size increase. For instance, Bosnian-English (bs-en) augmented with a cross-lingual Croatian-English (hr-en) datastore, which is only 60% of the size of Russian-English (ru-en), leads to +1.0 BLEU compared to using Russian-English. We conclude

---

[3] $\rho = 0.88$ with $p < 0.001$ for Slavic language grouping.

that it is difficult to predict which cross-lingual datastore will perform best.

In contrast, for high-resource languages it is *always* a better choice to use the bilingual datastore, even when larger cross-lingual datastores are available. For instance, Serbian-English (sr-en) with Serbian-English datastore of 3.6M tokens performs better (35.9 BLEU) than Serbian-English with the substantially larger cross-lingual datastore Russian-English (5.6M tokens, 34.6 BLEU).

When considering cross-lingual datastores that come from a more distant language family, using a bilingual datastore leads to better results, even for low-resource languages . Considering Georgian-English (ka-en), the bilingual datastore improvement (+3.9 BLEU) is larger than the improvement for its best cross-lingual datastore Greek-English (el-en, +2.1 BLEU).[4]

**Multilingual datastores perform best**   Since it is unclear which cross-lingual datastore performs best, we use as many languages as possible as a first attempt. This results in our largest datastore $\mathcal{D}_{(\text{ALL,en})}$, which has 125M entries. For almost all languages, except Russian-English (ru-en), this leads to better results than bilingual datastores. Low-resource languages show the largest improvements, where Bosnian-English (bs-en) has the largest improvement of +3.6 BLEU compared to bs-en datastore, or +0.7 BLEU compared to the best cross-lingual datastore. A problem for $\mathcal{D}_{(\text{ALL,en})}$ is slow inference speed, because $k$NN lookup in a large datastore is expensive. When decreasing the datastore size by focusing on bridge languages, we can construct a smaller datastore of size 86.4M, but its results are worse in almost all cases which is clearly reflected in the average scores. Finally, we consider a datastore that is constructed using linguistic similarity. It consists of languages from the same language grouping. We observe that this multilingual datastore is on par with the largest one, while significantly smaller, which is clearly reflected in the averages. For some languages, such as Belarusian-English (be-en) and Bosnian-English (bs-en), this even brings improvements of +0.6 BLEU and +0.5 BLEU compared to using $\mathcal{D}_{(\text{ALL,en})}$. We emphasize that multilingual

| $\mathcal{D}$ | $|\mathbb{T}|$ | $\mathbb{T}_{\text{be}}$ | $A\mathbb{T}_{\text{be}}$ | $\Delta$ BLEU |
|---|---|---|---|---|
| $\mathcal{D}_{(\text{bs,en})}$ | 23K | **20.5** | 20.4 | −0.1 |
| $\mathcal{D}_{(\text{sl,en})}$ | 95K | 20.9 | **21.3** | 0.4 |
| $\mathcal{D}_{(\text{mk,en})}$ | 73K | 20.8 | **21.2** | 0.4 |
| $\mathcal{D}_{(\text{sk,en})}$ | 202K | 21.9 | **21.3** | 0.3 |
| $\mathcal{D}_{(\text{cs,en})}$ | 305K | 21.5 | **21.7** | 0.2 |
| $\mathcal{D}_{(\text{uk,en})}$ | 347K | **22.4** | 22.2 | −0.2 |
| $\mathcal{D}_{(\text{hr,en})}$ | 359K | 21.4 | **21.7** | 0.3 |
| $\mathcal{D}_{(\text{sr,en})}$ | 417K | 21.3 | **21.8** | 0.5 |
| $\mathcal{D}_{(\text{bg,en})}$ | 431K | 21.3 | **21.8** | 0.5 |
| $\mathcal{D}_{(\text{pl,en})}$ | 421K | 21.3 | **22.1** | 0.8 |
| $\mathcal{D}_{(\text{ru,en})}$ | 533K | 21.7 | **22.3** | 0.6 |
| avg | 291K | 21.3 | **21.6** | 0.3 |
| $\mathcal{D}_{(\text{LG,en})}$ | − | 23.1 | **23.4** | 0.3 |

Table 2: be→en BLEU scores for Slavic grouping, with cross-lingual mapping ($A\mathbb{T}_{\text{be}}$) and without ($\mathbb{T}_{\text{be}}$). Training data size for mapping is shown as $|\mathbb{T}|$. Best results shown in **bold**.

datastores lead to best results for *all* languages we tested, including higher-resource directions such as Polish-English, (pl-en, +0.4 BLEU compared to bilingual) and Ukrainian-English (uk-en, +0.8 BLEU).

**Effectiveness of cross-lingual mapping**   We created a cross-lingual mapping from Belarusian (be) to other languages in the Slavic language grouping. We also created the inverse mapping, and constructed a Slavic language grouping datastore mapped to Belarusian representations.

Results are presented in Table 2. We observe that generally, Belarusian-English (be-en) performance is improved, especially for larger cross-lingual datastores such as Polish-English (pl-en, +0.8 BLEU). For bs-en and uk-en the mapping results in a slight quality decrease.[5]

### 4.4   Analysis

**Which languages are used?**   We explore the language origin of $k$NN-MT target token suggestions when using $\mathcal{D}_{(\text{ALL,en})}$ to augment the Norwegian-English (no-en) translation direction. The top 15 origins with highest probability mass are shown in Table 3. Surprisingly, despite consisting of only 5 out of 51 languages, the Germanic language group accounts for 23.2% of the suggestions. This helps to explain why $\mathcal{D}_{(\text{LG,en})}$ performs on par with $\mathcal{D}_{(\text{ALL,en})}$, even though it is more than ten times smaller. Full results are in Appendix B.

**Multilingual datastore speed**   Using the smaller datastore $\mathcal{D}_{(\text{LG,en})}$ results in significantly faster de-

---

[4]This is likely because the Greek language grouping combines different families: Greek (el) is a Hellenic language, whereas Georgian (ka) is from the Kartvelian language family. Therefore, their representations likely have larger differences than these of Bengali-English (be-en) and Ukrainian-English (uk-en), which come from the same family.

[5]This can possibly be explained by the small data size (for bs-en), and because uk-en and be-en are already relatively well aligned, since uk-en is the best cross-lingual datastore for be-en (see Table 4).

| $\mathcal{D}$ | $|\mathcal{D}|$ | $P_{\text{obs}}$ | $P_{\text{uni}}$ |
|---|---|---|---|
| **no-en** | 411K | 6.05% | 0.34% |
| it-en | 5.5M | 5.74% | 4.62% |
| fr-en | 5.1M | 5.62% | 4.29% |
| **nl-en** | 4.9M | 5.47% | 4.06% |
| es-en | 5.2M | 5.31% | 4.38% |
| **de-en** | 4.7M | 4.92% | 3.73% |
| he-en | 5.7M | 4.88% | 4.80% |
| bg-en | 4.7M | 4.77% | 3.95% |
| ro-en | 4.8M | 4.40% | 4.05% |
| **da-en** | 1.2M | 3.64% | 0.99% |
| el-en | 3.5M | 3.48% | 2.96% |
| hr-en | 3.3M | 3.15% | 2.73% |
| ru-en | 5.6M | 3.15% | 4.71% |
| sr-en | 3.6M | 3.11% | 3.02% |
| **sv-en** | 1.4M | 3.08% | 1.18% |

Table 3: Bilingual origins for the 15 datastore languages with the highest occurrence when augmenting Norwegian-English (no-en) with multilingual datastore $\mathcal{D}_{(\text{ALL,en})}$ datastore. $\mathcal{D}$ denotes bilingual datastore, and $|\mathcal{D}|$ the corresponding size. $P_{\text{obs}}$ are the observed origin percentages when decoding on the no-en test set. $P_{\text{uni}}$ are the uniform origin percentages, when taking into account the bilingual datastore sizes. **Bold** datastores indicate that they are from the no-en language grouping, and underlined means they are a bridge language. Darker colors indicate more probability mass.

coding speeds of up to 5.3x for Belarusian-English (be-en) compared to using $\mathcal{D}_{(\text{ALL,en})}$. More results are in Appendix C.

## 5   Conclusion

We have proposed a simple and effective approach to enhance quality for low-resource languages in *k*NN-MT. We augmented an mNMT model with cross-lingual and multilingual datastores of related and unrelated languages. We show that using multilingual datastores substantially improves translation quality for low-resource languages, while high-resource languages also improve. We find that by harnessing linguistic similarity, we can limit multilingual datastore size while preserving quality and significantly increasing inference speed. Finally, we show that by further aligning multilingual representations, we can more effectively use cross-lingual and multilingual datastores.

## Acknowledgements

This research was funded in part by the Netherlands Organization for Scientific Research (NWO) under project numbers VI.C.192.080 and VI.Veni.212.228. We thank Ali Araabi, Vlad Niculae, Yan Meng, Shaomu Tan, Ke Tran, Sony Trenous and Di Wu for their helpful suggestions and insights.

## Limitations

We use an mNMT model that is non-English-centric, which may present limitations in the generalizability of our multilingual *k*NN-MT results to other multilingual settings such as English-centric.

A limitation of *k*NN-MT, which has become the central focus of most subsequent *k*NN-MT research, is the steep increase in decoding time introduced by *k*NN-MT, as each decoding step requires a computationally expensive nearest neighbor search (Zheng et al., 2021; Martins et al., 2022a,b; Yang et al., 2022; Meng et al., 2022; Jiang et al., 2022).

While we built multilingual datastores using up to 51 languages, we did not investigate to what extent even larger multilingual datastores can further improve performance.

## Broader Impact

Machine translation poses potential risks, such as errors in translation. This risk is particularly high for low-resource languages. Adding *k*NN-MT likely decreases this risk, since translation quality improves. Using multilingual datastores likely further decreases this risk.

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

## A  Datastore information

We list information about all 51 datastores in Table 4. Note that often, language family and language grouping are consistent, i.e., all languages in a language grouping are from the same language family. However, there are some exceptions such as the language grouping Greek, which includes languages from the Hellenic, Kartvelian, Armenian, and Albanian families.

## B  Multilingual datastore language origins

We show language origins for the no-en translation direction augmented with a multilingual datastore consisting of all 51 languages in Table 5. We track the bilingual datastore origin of target token suggestions from *k*NN-MT during inference on the test set, and calculate the percentage based on the total number of suggested tokens. It should be noted that not all target token suggestions from *k*NN-MT are included in the generated target sentence.

From Table 5 we observe that the distribution of observed language origins generally follows a uniform distribution based on bilingual datastore size. The largest outlier is the no-en datastore, which is used for $6.05\%$ of the generations, as opposed to the $0.34\%$ that would be expected from a uniform distribution.

Additionally, we find that all five languages from the Germanic datastore, to which no-en belongs, are oversampled compared to the uniform distribution. This set of just five languages is responsible for $23.2\%$ of the matches.

Furthermore, we observe that datastores from several bridge languages, including ar-en, pl-en, hu-en, ko-en and ja-en, are undersampled compared to the uniform expectation. This discrepancy can likely be attributed to the fact that these languages are relatively distant from Norwegian, resulting in dissimilar representations.

## C  Multilingual datastore inference speed

We present multilingual datastore inference speeds in Figure 1. We set $k$ to $64$, and present results for all multilingual datastores, using a single source language into English for each language grouping. We average results over 3 runs. We observe a clear trend: smaller datastores result in substantially faster decoding times. For be-en, using the Language Grouping instead of the All multilingual datastore leads to a $5.3$x speed improvement. For no-en and ka-en, the improvements are $3.0$x and $2.6$x. In terms of quality, be-en improves when using the Language Grouping datastore ($+0.6$ BLEU), while no-en and ka-en have similar performance ($-0.1$ BLEU and $+0.0$ BLEU).

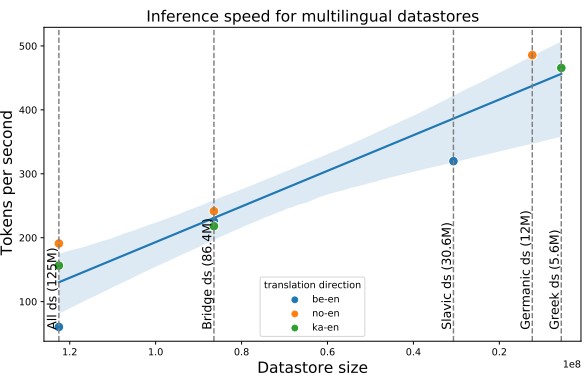

Figure 1: Inference speed for Greek, Germanic, Slavic, Bridge, and All multilingual datastores. The x-axis displays the datastore size (large to small), and the y-axis shows the corresponding tokens per second. A clear linear trend can be observed: smaller datastores result in substantially faster decoding times.

| datastore | size | family | grouping | script | bridge | M2M100 | +kNN-MT | diff |
|---|---|---|---|---|---|---|---|---|
| **kk-en** | 84K | Turkic | Turkic | Cyrillic | | 2.1 | 2.6 | 0.5 |
| **be-en** | 116K | Slavic | Slavic | Cyrillic | | 19.2 | 20.9 | 1.7 |
| **bn-en** | 127K | Indo-Aryan | Indo | Eastern-Nagari | ✓ | 9.3 | 13.9 | 4.6 |
| **ms-en** | 132K | Malayo-Polyn. | Malayo | Latin | | 28.8 | 30.6 | 1.8 |
| **bs-en** | 146K | Slavic | Slavic | Latin | | 31.5 | 33.1 | 1.6 |
| **az-en** | 153K | Turkic | Turkic | Cyrillic | | 8.8 | 10.0 | 1.2 |
| **ta-en** | 156K | Dravidian | Indo | Tamil | ✓ | 0.4 | 0.9 | 0.5 |
| **ur-en** | 158K | Indo-Aryan | Indo | Arabic | | 14.4 | 16.9 | 2.5 |
| **mn-en** | 181K | Mongolic | Mongolic | Cyrillic | | 5.1 | 7.1 | 2.0 |
| **mr-en** | 241K | Indo-Aryan | Indo | Devanagari | | 3.9 | 6.2 | 2.3 |
| **gl-en** | 254K | Romance | Romance | Latin | | 32.4 | 34.0 | 1.6 |
| **et-en** | 280K | Uralic | Uralic | Latin | | 23.5 | 25.3 | 1.8 |
| **ka-en** | 332K | Kartvelian | Greek | Georgian | | 10.8 | 14.7 | 3.9 |
| **no-en** | 411K | Germanic | Germanic | Latin | | 42.8 | 45.6 | 2.8 |
| **hi-en** | 481K | Indo-Aryan | Indo | Devanagari | ✓ | 17.9 | 23.3 | 5.4 |
| **sl-en** | 520K | Slavic | Slavic | Latin | | 24.9 | 27.3 | 2.4 |
| **hy-en** | 544K | Armeian | Greek | Armenian | | 16.8 | 20.1 | 3.3 |
| **my-en** | 558K | Sino-Tibetan | Mongolic | Burmese | | 0.4 | 1.5 | 1.1 |
| **fi-en** | 623K | Uralic | Uralic | Latin | ✓ | 21.0 | 22.8 | 1.8 |
| **mk-en** | 683K | Slavic | Slavic | Cyrillic | | 29.3 | 32.8 | 3.5 |
| **lt-en** | 1.1M | Baltic | Uralic | Latin | ✓ | 24.7 | 28.2 | 3.5 |
| **sq-en** | 1.2M | Albanian | Greek | Latin | | 31.9 | 35.8 | 3.9 |
| **da-en** | 1.2M | Germanic | Germanic | Latin | | 40.0 | 44.5 | 4.5 |
| **pt-en** | 1.2M | Romance | Romance | Latin | ✓ | 38.9 | 42.0 | 3.1 |
| **sv-en** | 1.4M | Germanic | Germanic | Latin | ✓ | 37.3 | 41.0 | 3.7 |
| **sk-en** | 1.6M | Slavic | Slavic | Latin | | 28.4 | 32.6 | 4.2 |
| **id-en** | 2.3M | Malayo-Polyn. | Malayo | Latin | ✓ | 29.1 | 32.5 | 3.4 |
| **th-en** | 2.6M | Kra-Dai | Mongolic | Thai | | 2.3 | 8.5 | 6.2 |
| **cs-en** | 2.7M | Slavic | Slavic | Latin | | 27.5 | 31.4 | 3.9 |
| **uk-en** | 2.9M | Slavic | Slavic | Cyrillic | | 24.7 | 29.1 | 4.4 |
| **hr-en** | 3.3M | Slavic | Slavic | Latin | | 32.2 | 37.0 | 4.8 |
| **el-en** | 3.5M | Hellenic | Greek | Greek | ✓ | 32.6 | 38.3 | 5.7 |
| **sr-en** | 3.6M | Slavic | Slavic | Cyrillic | | 30.7 | 35.9 | 5.2 |
| **hu-en** | 3.9M | Uralic | Uralic | Latin | ✓ | 23.3 | 26.8 | 3.5 |
| **fa-en** | 4.0M | Iranian | Arabic | Arabic | ✓ | 22.7 | 27.6 | 4.9 |
| **de-en** | 4.5M | Germanic | Germanic | Latin | ✓ | 31.7 | 36.9 | 5.2 |
| **vi-en** | 4.6M | Vietic | Chinese | Latin | ✓ | 23.7 | 27.2 | 3.5 |
| **bg-en** | 4.7M | Slavic | Slavic | Cyrillic | | 34.4 | 39.5 | 5.1 |
| **pl-en** | 4.7M | Slavic | Slavic | Latin | ✓ | 21.1 | 25.0 | 3.9 |
| **ro-en** | 4.8M | Romance | Romance | Latin | | 30.6 | 35.4 | 4.8 |
| **nl-en** | 4.9M | Germanic | Germanic | Latin | ✓ | 31.9 | 36.2 | 4.3 |
| **tr-en** | 4.9M | Turkic | Turkic | Latin | ✓ | 22.4 | 26.5 | 4.1 |
| **fr-en** | 5.1M | Romance | Romance | Latin | ✓ | 35.1 | 40.3 | 5.2 |
| **es-en** | 5.2M | Romance | Romance | Latin | ✓ | 36.6 | 41.9 | 5.3 |
| **zh-en** | 5.4M | Chinese | Chinese | Chinese | ✓ | 16.3 | 20.2 | 3.9 |
| **ja-en** | 5.5M | Japonic | Chinese | Kanji | ✓ | 10.6 | 13.5 | 2.9 |
| **it-en** | 5.5M | Romance | Romance | Latin | | 33.4 | 38.4 | 5.0 |
| **ko-en** | 5.5M | Koreanic | Chinese | Hangul | ✓ | 16.2 | 19.5 | 3.3 |
| **ru-en** | 5.6M | Slavic | Slavic | Cyrillic | ✓ | 21.6 | 25.8 | 4.2 |
| **he-en** | 5.7M | Semitic | Arabic | Hebrew | ✓ | 31.2 | 36.8 | 5.6 |
| **ar-en** | 5.8M | Arabic | Arabic | Arabic | ✓ | 26.2 | 31.4 | 5.2 |
| **average** | 2.5M | – | – | 23.4 | 27.0 | 3.6 | | |
| **total** | 125M | – | – | – | – | – | | |

Table 4: Datastore information for all 51 languages into English that we use. Column grouping indicates language grouping from M2M100. Column bridge indicates whether the language is a bridge language in M2M100. The three final rows show BLEU scores for the base model (M2M100), augmented with *k*NN-MT (+*k*NN-MT), and their difference (diff).

| $\mathcal{D}$ | $|\mathcal{D}|$ | $P_{\text{obs}}$ | $P_{\text{uni}}$ |
|---|---|---|---|
| **no-en** | 411K | 6.05% | 0.34% |
| it-en | 5.5M | 5.74% | 4.62% |
| fr-en | 5.1M | 5.62% | 4.29% |
| **nl-en** | 4.9M | 5.47% | 4.06% |
| es-en | 5.2M | 5.31% | 4.38% |
| **de-en** | 4.7M | 4.92% | 3.73% |
| he-en | 5.7M | 4.88% | 4.80% |
| bg-en | 4.7M | 4.77% | 3.95% |
| ro-en | 4.8M | 4.40% | 4.05% |
| **da-en** | 1.2M | 3.64% | 0.99% |
| el-en | 3.5M | 3.48% | 2.96% |
| hr-en | 3.3M | 3.15% | 2.73% |
| ru-en | 5.6M | 3.15% | 4.71% |
| sr-en | 3.6M | 3.11% | 3.02% |
| **sv-en** | 1.4M | 3.08% | 1.18% |
| vi-en | 4.6M | 3.06% | 3.84% |
| ar-en | 5.8M | 3.05% | 4.87% |
| pl-en | 4.7M | 2.60% | 3.95% |
| cs-en | 2.7M | 2.35% | 2.27% |
| hu-en | 3.9M | 2.34% | 3.28% |
| tr-en | 4.9M | 2.21% | 4.07% |
| id-en | 2.3M | 2.11% | 1.90% |
| fa-en | 4.1M | 1.93% | 3.39% |
| pt-en | 1.2M | 1.81% | 1.03% |
| zh-en | 5.4M | 1.67% | 4.50% |
| ko-en | 5.5M | 1.66% | 4.63% |
| sk-en | 1.6M | 1.48% | 1.36% |
| ja-en | 5.5M | 1.17% | 4.60% |
| sq-en | 1.2M | 1.00% | 0.97% |
| mk-en | 683K | 0.83% | 0.57% |
| lt-en | 1.1M | 0.78% | 0.91% |
| sl-en | 520K | 0.55% | 0.44% |
| fi-en | 623K | 0.48% | 0.52% |
| th-en | 2.6M | 0.36% | 2.20% |
| gl-en | 254K | 0.31% | 0.21% |
| hy-en | 544K | 0.26% | 0.45% |
| et-en | 280K | 0.20% | 0.23% |
| hi-en | 481K | 0.19% | 0.40% |
| bs-en | 146K | 0.14% | 0.12% |
| ka-en | 332K | 0.14% | 0.28% |
| my-en | 558K | 0.12% | 0.47% |
| ms-en | 132K | 0.08% | 0.11% |
| mr-en | 241K | 0.07% | 0.20% |
| be-en | 116K | 0.07% | 0.10% |
| ur-en | 158K | 0.05% | 0.13% |
| bn-en | 127K | 0.04% | 0.11% |
| mn-en | 181K | 0.04% | 0.15% |
| ta-en | 156K | 0.03% | 0.13% |
| az-en | 153K | 0.03% | 0.13% |
| kk-en | 84K | 0.02% | 0.07% |

Table 5: Bilingual origins when augmenting no-en with multilingual datastore $\mathcal{D}_{(\text{ALL,en})}$ datastore. $\mathcal{D}$ denotes bilingual datastore, and $|\mathcal{D}|$ the corresponding size. $P_{\text{obs}}$ are the observed origin percentages when decoding on the no-en test set. $P_{\text{uni}}$ are the uniform origin percentages, when taking into account the bilingual datastore sizes. **Bold** datastores indicate that they are from the no-en language grouping, and underlined means they are a bridge language. Darker colors indicate more probability mass.