# OpenReview forum: "Multilingual \textit{k}-Nearest-Neighbor Machine Translation"
_EMNLP/2023/Conference — EMNLP 2023 Main_

### Official Review · Reviewer_aszQ · 2023-07-30

**Soundness:** 4

**Excitement:**

3: Ambivalent: It has merits (e.g., it reports state-of-the-art results, the idea is nice), but there are key weaknesses (e.g., it describes incremental work), and it can significantly benefit from another round of revision. However, I won't object to accepting it if my co-reviewers champion it.

**Paper Topic And Main Contributions:**

This paper tries to combine multilingual NMT with kNN-based NMT to achieve better translation quality for low-resource languages.

Major contributions:
- proposed an approach to augment a multilingual NMT model with datastores of related and unrelated languages;
- showed the approach can improve low-resource translation quality;
- a tradeoff between quality and speed;

**Questions For The Authors:**

- the paper only shows xx->en results which might be different from en->xx results, so it would be good to also show some results on en->xx. Could you also add en->xx results too?

**Reasons To Accept:**

- this paper should have enough contents and results for a short paper;
- the improvements on low-resource languages look good in BLEU scores;
- the experiments are comprehensive;
- the paper is well written;

**Reasons To Reject:**

- the proposed approach is a bit complicated, as it combines multilingual NMT with kNN-based NMT, so it might be hard for other people to follow;
- the paper only shows xx->en results which might be different from en->xx results, so it would be good to also show some results on en->xx;

**Reproducibility:**

3: Could reproduce the results with some difficulty. The settings of parameters are underspecified or subjectively determined; the training/evaluation data are not widely available.

**Reviewer Confidence:**

4: Quite sure. I tried to check the important points carefully. It's unlikely, though conceivable, that I missed something that should affect my ratings.

**Typos Grammar Style And Presentation Improvements:**

- Line 240: "a first attempt" should be "the first attempt";

---

> ### Author Rebuttal · Authors · 2023-08-28
>
> Thank you for your review!
>
> * the proposed approach is a bit complicated, as it combines multilingual NMT with kNN-based NMT, so it might be hard for other people to follow;
>
> Regarding your comment that our approach seems a bit complicated; we respectfully disagree. We agree with Reviewer oGvF and Reviewer oA3E that our idea is clear and straightforward, resulting in a simple yet effective method. A multilingual NMT system is a single system that can serve a large number of languages. Adding kNN is an extra step, which requires the construction of datastores. In contrast to [1], our approach allows us to use a *single multilingual datastore* for a large number of languages, which is much simpler than creating *separate bilingual datastores* per language direction. Note that we will release our code (which is based on the commonly used fairseq framework) to the community, which makes reproducing results and building on our approach very simple. In addition, we will expand Section 2 (i.e., the section that explains kNN-MT) when our paper gets accepted, since we will get additional space. We hope this change will make our paper even easier to follow.
>
> * the paper only shows xx->en results which might be different from en->xx results, so it would be good to also show some results on en->xx;
> * the paper only shows xx->en results which might be different from en->xx results, so it would be good to also show some results on en->xx. Could you also add en->xx results too?
>
> Regarding your question about showing en->xx results in addition to xx->en; this is a good point. We note that *k*NN-MT builds datastores consisting of source context (hidden representations) and target tokens. Therefore, creating cross-lingual/multilingual datastores that (partly) consist of target tokens that are from a different target language will not lead to good results. As an extreme example, if we create a multilingual datastore consisting of {En-Fr, En-Zh} data, the En-Zh portion is not beneficial for En-Fr translations because of lack of overlap. We will add some en->xx results plus this explanation in the updated version of the paper.

---

### Official Review · Reviewer_oA3E · 2023-08-01

**Soundness:** 4

**Excitement:**

4: Strong: This paper deepens the understanding of some phenomenon or lowers the barriers to an existing research direction.

**Paper Topic And Main Contributions:**

The paper proposes a multilingual method for kNN NMT. The author mixes datastores of different language pairs. And use a linear mapping method to make the representation closer.

**Questions For The Authors:**

Have you compared with a multilingual model?

**Reasons To Accept:**

1. The idea is clear and straightforward. The motivation is solid.
2. The experiments show significant improvements.
3. The detailed analysis is convincing.

**Reasons To Reject:**

1. The improvement of cross-lingual mapping is relatively small (0.3 BLEU). The contribution of this part is a little bit minor.
2. Similarly, the innovation is not strong enough.
3. There should be a baseline of a native multilingual datastore, that is, a multilingual model.

**Reproducibility:**

3: Could reproduce the results with some difficulty. The settings of parameters are underspecified or subjectively determined; the training/evaluation data are not widely available.

**Reviewer Confidence:**

5: Positive that my evaluation is correct. I read the paper very carefully and I am very familiar with related work.

---

> ### Author Rebuttal · Authors · 2023-08-28
>
> Thank you for your review!
>
> * The improvement of cross-lingual mapping is relatively small (0.3 BLEU). The contribution of this part is a little bit minor.
> * Similarly, the innovation is not strong enough.
>
> Regarding your comment that the innovation is not strong enough: our method is simple, which we see as a strength, and it shows significant improvements, like you mentioned. (This is without the additional improvement of the cross-lingual mapping, which brings an additional small benefit and is not the main result of our work.) Different from prior work, that augments a *multilingual* model with *bilingual datastores* [1] or uses a pre-trained multilingual model to create a *monolingual datastore* [2], we are the first to develop methods to carefully construct *multilingual datastores*, and we show that this significantly outperforms bilingual datastores. We further empirically show that multilingual datastore size can be limited while preserving quality compared to our largest multilingual datastore, which significantly increases inference speed.
>
> * There should be a baseline of a native multilingual datastore, that is, a multilingual model.
> * Have you compared with a multilingual model?
>
> Regarding your comment about the absence of a multilingual datastore / multilingual model baseline: this is incorrect, we *do* have these results in the paper, and we therefore disagree that this is a reason to reject. Please see Table 1 for these numbers: scores for the base multilingual model (i.e., the M2M100 model without datastore) are listed in the *first columns* (“base”), with datastore sizes of 0; scores for the base multilingual model + a complete multilingual datastore (i.e., the M2M100 model augmented with datastores consisting of all TED languages into English) are listed in the *last columns* (D_{ALL}), with datastore size of 125M. We discuss these results in Section 4.3. With regards to the baselines, we observe that using any kind of datastore always outperforms the base model (as one would expect). More importantly, we show that using our complete multilingual datastore (D_{ALL}) works better than bilingual (cross-lingual) datastores for all languages except Ru-En. We then show that datastores constructed using linguistic similarity (D_{LG}) allows for significantly smaller datastores (resulting in faster inference of up to 5.3x, see Figure 1 in Appendix), while being on par with the complete datastore in terms of BLEU.
>
> [1] Urvashi Khandelwal, Angela Fan, Dan Jurafsky, Luke Zettlemoyer and Mike Lewis. Nearest Neighbor Machine Translation. ICLR 2021.
>
> [2] Jiahuan Li, Shanbo Cheng, Zewei Sun, Mingxuan Wang and Shujian Huang. Better Datastore, Better Translation: Generating Datastores from Pre-Trained Models for Nearest Neural Machine Translation. arXiv 2022.

---

### Official Review · Reviewer_oGvF · 2023-08-04

**Soundness:** 4

**Excitement:**

3: Ambivalent: It has merits (e.g., it reports state-of-the-art results, the idea is nice), but there are key weaknesses (e.g., it describes incremental work), and it can significantly benefit from another round of revision. However, I won't object to accepting it if my co-reviewers champion it.

**Paper Topic And Main Contributions:**

The paper explores multilingual k-nearest neighbor (KNN) approach to machine translation (MT), with focus on low resource languages. The authors compare multiple datastore types for KNN, including bilingual and multilingual ones with different selection of language groups. Results on TED show large improvements on low resource languages when using the proposed multilingual datastore as opposed to the bilingual ones, in addition to smaller improvements for high resource languages. The authors also provide reliable language group selection to reduce the datastore size and report latency analysis of the inference stage.

**Reasons To Accept:**

* The paper is well written, the method proposed of multilingual datastore is simple yet effective
* The method results in substantial improvements on TED across different languages
* Latency analysis is provided
* Additional meaningful analysis such as which datastore are mostly used is also provided
* The authors mention they will release the code upon publication

**Reasons To Reject:**

* The contribution can be deemed minor, previous work used similar methods but did not focus on low resource languages

**Reproducibility:**

5: Could easily reproduce the results.

**Reviewer Confidence:**

4: Quite sure. I tried to check the important points carefully. It's unlikely, though conceivable, that I missed something that should affect my ratings.

**Typos Grammar Style And Presentation Improvements:**

* lines 116,129 issue in spacing
* Table 1: build -> built

---

> ### Author Rebuttal · Authors · 2023-08-28
>
> Thank you for your review!
>
> * The contribution can be deemed minor, previous work used similar methods but did not focus on low resource languages
>
> Our work is different from previous work in important ways, and the difference is not limited to our focus on low-resource languages. There is related work that combines *k*NN-MT with multilingual models: [1] augments a multilingual *model* with *bilingual datastores*; [2] uses a pre-trained multilingual model to create a *monolingual datastore* of the target language. In contrast, we develop methods to carefully construct *multilingual datastores*, which we show outperforms bilingual datastores. Our method is simple, which we see as a strength. The method shows strong improvements for low-resource languages (and smaller improvements for high-resource languages) compared to strong baselines.
>
> We’ll make sure to fix the typo and the issues with spacing, thanks for pointing it out.
>
> [1] Urvashi Khandelwal, Angela Fan, Dan Jurafsky, Luke Zettlemoyer and Mike Lewis. Nearest Neighbor Machine Translation. ICLR 2021.
>
> [2] Jiahuan Li, Shanbo Cheng, Zewei Sun, Mingxuan Wang and Shujian Huang. Better Datastore, Better Translation: Generating Datastores from Pre-Trained Models for Nearest Neural Machine Translation. arXiv 2022.

---

### Meta-Review · Area_Chair_KELB · 2023-09-21

**Recommendation:** 4

**Metareview:**

The paper proposes using kNN-MT for low-resource languages. Most prominently, it combines representations from multiple different languages into the same datastore --- traditional kNN-MT uses huge datastores built for one language. The work contains extensive experimentation and reviewers are in consensus for acceptance.

---

### Decision · Program_Chairs · 2023-10-07

**Decision:**

Accept-Main

**Comment:**

The paper proposes using kNN-MT for low-resource languages. Most prominently, it combines representations from multiple different languages into the same datastore --- traditional kNN-MT uses huge datastores built for one language. The work contains extensive experimentation and reviewers are in consensus for acceptance.